# Joint analysis of expression levels and histological images identifies genes associated with tissue morphology

Jordan T. Ash[1,4], Gregory Darnell[2,4], Daniel Munro [2,4] & Barbara E. Engelhardt [1,3 ✉]

Histopathological images are used to characterize complex phenotypes such as tumor stage. Our goal is to associate features of stained tissue images with high-dimensional genomic markers. We use convolutional autoencoders and sparse canonical correlation analysis (CCA) on paired histological images and bulk gene expression to identify subsets of genes whose expression levels in a tissue sample correlate with subsets of morphological features from the corresponding sample image. We apply our approach, ImageCCA, to two TCGA data sets, and find gene sets associated with the structure of the extracellular matrix and cell wall infrastructure, implicating uncharacterized genes in extracellular processes. We find sets of genes associated with specific cell types, including neuronal cells and cells of the immune system. We apply ImageCCA to the GTEx v6 data, and find image features that capture population variation in thyroid and in colon tissues associated with genetic variants (image morphology QTLs, or imQTLs), suggesting that genetic variation regulates population variation in tissue morphological traits.

[1] Department of Computer Science, Princeton University, Princeton, NJ, USA. [2] Lewis-Sigler Institute for Integrative Genomics, Princeton University, Princeton, NJ, USA. [3] Center for Statistics and Machine Learning, Princeton University, Princeton, NJ, USA. [4]These authors contributed equally: Jordan T. Ash, Gregory Darnell, Daniel Munro. ✉email: bee@princeton.edu

Histological and histopathological images—high-resolution microscopic images of healthy or diseased tissue samples that have been sectioned and stained—are essential for identifying and characterizing complex histopathological phenotypes. Pathologists study tissues using stained imaging for scientific research on cellular morphology and tissue structure and for medical practice. For example, visual inspection of biopsied tissue is a major component of cancer diagnosis, since cancer is known to affect the morphological properties of tissues, including extracellular structure and cell size, shape, and organization[1].

There has been considerable research in computationally analyzing pathological image data to develop automated cancer diagnoses. Earlier approaches typically involved the extraction of predetermined morphological, textural, and fractal image features from histological images[2]. The resulting image feature vectors then are used to classify the pathological status of the sample[3,4]. Because this feature extraction relies on human-defined features, challenges arise as a result of cross-tumor heterogeneity and the variance inherent in histology and pathology[5]. Supervised image classification, often using deep learning methods, has also been effective for immunohistochemical-stained tissue samples, allowing biomarkers of interest to be used to classify tumor status, for example[6–8].

Complementary to visual inspection of histological images, gene expression is used to study cellular activity on the molecular level. Bulk gene expression data have been used to characterize and understand cellular differences between sample tissues[9], disease phenotypes[10], environments[11], or exposures[12]. Current work has mainly focused on finding genotypes and gene expression levels associated with disease phenotypes[9,13]. Single cell imaging studies have begun to study the connection between expression and cellular morphology[14,15], but throughput, number of transcripts imaged, number of cells, and image analysis pose challenges to this technology as an all-purpose solution. More generally, analyses to identify sets of genes whose expression levels are correlated with cellular physiology and tissue phenotypes will enable investigation into both basic cellular biology and drivers of cellular morphology associated with disease. Here, we are interested in identifying genes and genotypes associated with quantitative phenotypes derived from stained images of tissue sections (H&E stains), which may be used as informative endophenotypes.

Association studies, rather than predictive and diagnostic studies, involving histological image data have not been broadly undertaken, despite their importance. This due to three challenges. First, histological samples paired with genomic observations on the same (adjacent tissue) samples are rare outside of cancer studies. Second, it is not clear how to identify biologically relevant features automatically from histological images. Previous work on this subject involved extracting hand-engineered features from images and computing pairwise correlations with gene expression data[16]. Methods exist to analyze histological images automatically, but often these methods extract image features that are not associated with genomic features or disease status[17]. Third, assuming that image features are available, univariate tests for correlation between genomic and image features are often confounded by technical and biological covariates including image scale and the time from sample collection to processing. It is not clear how to control for large effect confounders when relevant biological signal may also be reflected in genes and image features.

In this work, we address the three technical difficulties in a framework called ImageCCA. We automatically extract image features using a machine learning technique called a convolutional autoencoder (CAE)[18]. A CAE is an unsupervised deep learning method that produces a small set of numeric features

characterizing each input image that allows the reconstruction of the input images with minimal loss[18]. These image feature representations are intended to capture variance in the image as a whole, but also find image features that are predictive of class labels, such as tumor versus healthy samples or tissue type.

We address the issue of controlling for technical and biological confounding in these associations by using sparse canonical correlation analysis (CCA) to partition the variation in the samples by identifying correlated sets of genes and histological image features. Probabilistic CCA finds linear mappings for two sets of observation from paired samples onto a shared low-dimensional space; this low-dimensional space is the one for which the two observation types are maximally correlated with each other[19,20]. Because these linear mappings involve tens of thousands of genes when applied to genome-wide gene expression data from human samples, we use a sparse form of CCA to find small subsets of genes and image features whose values correlate most strongly with each other[21]. CCA can be thought of as jointly modeling and partitioning the contributors to variance in the gene expression levels and image features, including technical and biological covariates, and biological signals. Thus, a single CCA component—capturing variation in the samples due to a subset of genes and image features—implicitly captures variation specific to that feature subset, controlling for variation due to confounding and other signals captured in the other components. We interpret the variation captured in the CCA components by examining the enriched molecular functions and tissue specificity of the genes in each component, and also examining the cellular morphology of the images differentiated by that component.

This paper proceeds as follows. First, we give a motivated overview of ImageCCA for the joint analysis of paired gene expression and histological image data. Next, we apply this framework to three data sets with histological images and gene expression levels on paired samples. We demonstrate the biological significance of the resulting associations using functional analyses of the subsets of genes that correlate with image features using enrichment studies. Finally, we present genotype associations with specific image features—image morphology QTLs—that drive population variation in histological image morphology via specific genes.

## Results

**ImageCCA for gene-image associations**. In order to study associations between cellular morphology and gene expression levels, we developed a framework, ImageCCA, to correlate automatically extracted image features from histological images with paired gene expression; see the "Methods" section for complete details. We applied two variants of our method, an unsupervised and a supervised version, to three different studies that include histological images and bulk RNA-sequencing data from paired tissue samples.

**Tissue sample data sets**. First, we applied our method to data from the Cancer Genome Atlas (TCGA) Breast Invasive Carcinoma (BRCA) study[22]. We used 1541 histological images from 1106 tissue biopsy samples, taken from 1073 breast cancer patients. Of these, 1502 images were of 1073 primary tumor samples, seven images were of seven metastatic tumor samples, and 32 images were of 26 normal tissue samples. The bulk RNA-sequencing data for paired samples include quantifications of expression levels in transcripts per million (TPM) units for 20,501 genes. The primary and metastatic tumor samples were grouped into a single *tumor* class label, in contrast to a *normal* label, for the supervised version of our approach.

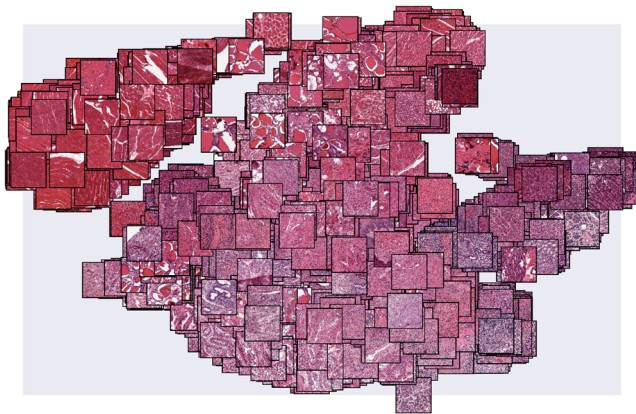

**Fig. 1 The embeddings of GTEx histological images.** The image feature representation estimated by ImageCCA for each of the GTEx histological images may be visualized by embedding the images based on their feature values into two dimensions using t-SNE[56]. We plot each histological image in this two-dimensional space. Images with similar morphological features are closer together, with skeletal muscle tissues forming a noticeably distinct cluster from the remaining tissue types in the upper left corner.

Next, we applied ImageCCA to samples from the TCGA brain lower grade glioma (LGG) study data, which includes both primary and recurrent tumor types[23]. These data include 484 histopathological images from 401 tissue biopsy samples taken from 392 LGG patients. Of these, 471 images were derived from 388 primary tumor samples, and 13 images were derived from 13 recurrent tumor samples. The bulk RNA-sequencing data for these samples include quantifications of expression levels in TPM units for 20,501 genes. The class labels used for supervised training were *primary tumor* and *recurrent tumor*.

Finally, we ran our method on data from the Genotype-Tissue Expression (GTEx) project[9]. These GTEx v6 data include bulk RNA-sequencing, genotypes, and histological images for each sample, across 29 types of non-diseased tissues. We used histological images and bulk gene expression data in transcripts per million (TPM) units from 2221 samples across 499 individuals. These bulk RNA-seq data included quantification of expression levels in TPM for 18,659 genes. The GTEx sample class label is the sample tissue type.

**Histological images and feature extraction.** Each of these three studies includes images of tissue slices fixed to slides and stained with hematoxylin and eosin (H&E). The CAE was used to embed these images into a 1024-dimensional space, as in state-of-the-art image CAEs[24] (Supplementary Fig. 1). Using the CAE in an unsupervised approach, the embedding is estimated with the objective of reconstructing the original image as accurately as possible, where the objective is minimizing the $\ell_2$ distance between the original and the reconstructed image, using only the estimated 1024 features. This low-dimensional representation encodes visual properties of the images without regard to cancer status or tissue type. In this feature space, images with similar morphological features tend to be closer to each other in Euclidean space, while images with dissimilar features tend to be farther apart (Fig. 1).

The feature representation from the CAE quantifies many types of histological variance, but we are often interested in the morphological differences between tissue types or pathological states. To find these differences, we added a multilayer perceptron (MLP) to the pre-trained encoding portion of the CAE, and trained the MLP to distinguish histological images according to the labels in the data set—tumor and normal tissue for BRCA and primary and recurrent tumor for LGG. The MLP network adds supervision to the feature extraction process: The encoder will identify image features that are useful for classification—for example, distinguishing morphological features of tumors versus healthy tissues—rather than for image reconstruction.

The supervised analysis results are shown as a proof of concept. However, because of the small numbers of images and unbalanced label classes, the supervised analysis did not produce results that differed from the unsupervised results substantially, and were harder to justify in the context of the downstream CCA. Unless stated explicitly, the results presented below represent the unsupervised application of the CAE.

**Gene expression and image components.** We applied sparse CCA to the 1024 image features from the unsupervised CAE and the paired-sample gene expression to find subsets of gene expression values that correlate to subsets of image features. Sparse CCA performs the same projection as CCA into a shared latent space, but zeros are encouraged in the projection matrix, identifying small numbers of genes and image features responsible for the variation captured in that component[25]. We calculated the first 100 CCA variables for each data set (see the "Methods" section). Each latent component estimated by the sparse CCA method includes (i) non-zero weights on a subset of image features, representing their contribution to the variance in the images; (ii) non-zero weights on a subset of genes correlated with those image features, representing the contribution of those genes to the gene expression variance; and (iii) a factor representing the contribution of these image and gene features to each of the $n$ paired samples.

We study and validate the patterns captured in each of these components in two ways. First, we study the subsets of non-zero genes and non-zero image features in each component. We perform Gene Ontology (GO) term enrichment tests, and find tissue types in which the set of genes is expressed, to understand the biological signal captured in a component. Second, we explore the images with the most extreme (positive and negative) values in the CCA factor corresponding to each component representing the two extremes of a linear ranking of samples with respect to a factor. This allows us to characterize the component's signal through exploring the visual differences in the images with the most extreme factor values. We confirm the cross-observation signals by permuting sample labels on one of the two observations and quantifying the difference in variance explained by the permuted and true CCA components.

**BRCA pathological image analysis.** We extracted 100 CCA components from the BRCA pathological image and gene expression data[22]. In the unsupervised setting, these components included an average of 255 nonzero genes and 90 nonzero image features. The proportion of variance explained (PVE) of the components shows non-monotone decay across the 100 components (Supplementary Fig. 4a). In the supervised setting, these components included an average of 802 nonzero genes and 4 nonzero image features. To validate the signal identified in CCA, we permuted the labels into three different ways on the gene expression values and reran CCA 10 times; we found that the variance captured by the true data was substantially larger than the variance captured by the permuted data, indicating that the CCA components capture meaningful latent structure among the two observation types (Supplementary Fig. 5).

We performed GO term enrichment tests with the subsets of genes for each BRCA sparse CCA component to interpret the signals captured in each component (Supplementary Tables 1 and 2). The top BRCA component was enriched for genes

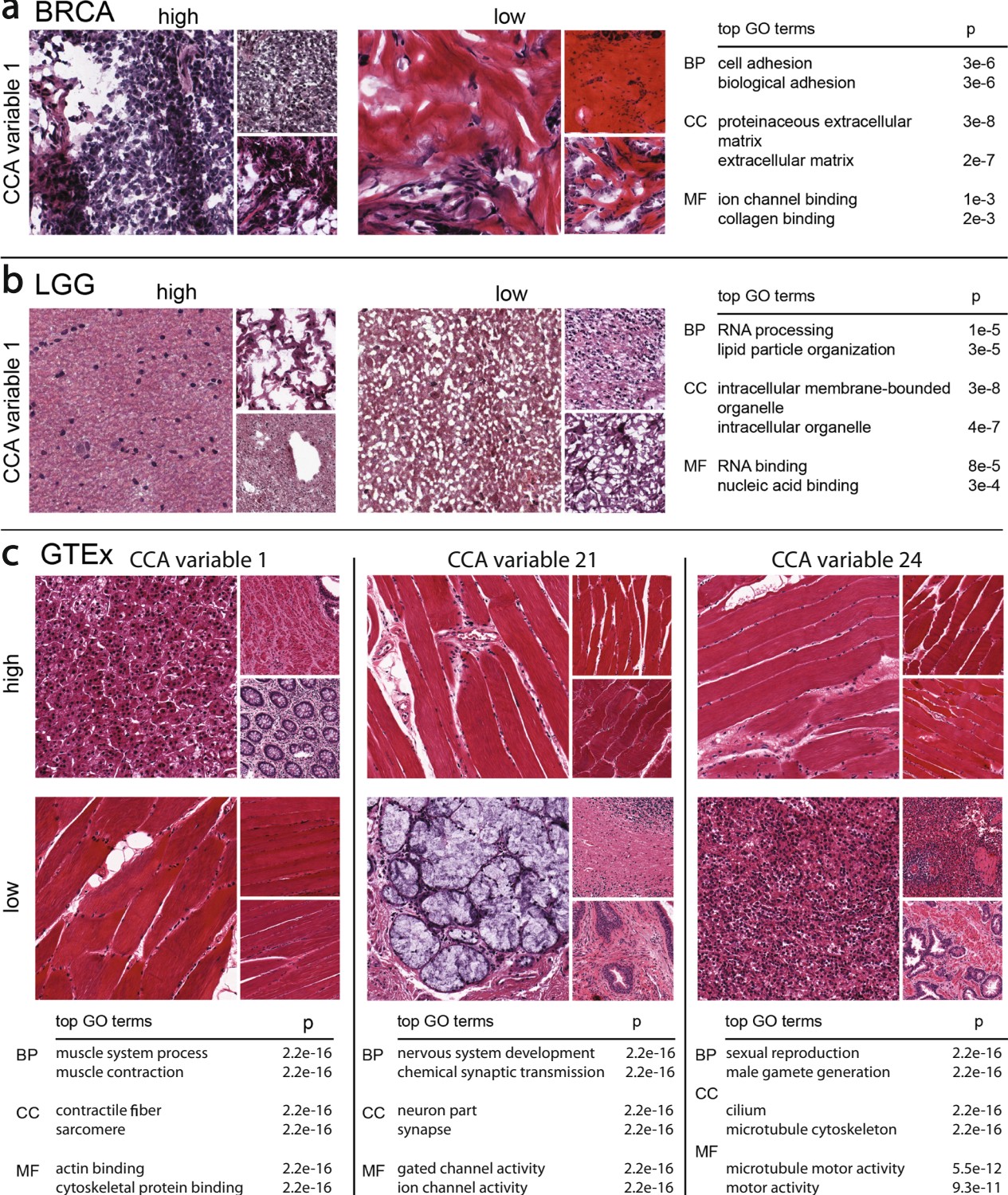

**Fig. 2 Results using ImageCCA for three different data sets.** We report images sampled from those with the most extreme magnitude positive and negative (10% and 90% in a linear ranking) CCA variable values, and top two GO terms that are most enriched with the corresponding genes with extreme loading values in the same component. BP biological process, CC cellular component, MF molecular function. The *p*-values reported are uncorrected Fisher's exact tests. Panel **a**: the first component of the BRCA ImageCCA results; Panel **b**: the first component of the LGG ImageCCA results; Panel **c**: three components of the GTEx ImageCCA results.

involved in *cell adhesion* ($p \le 3 \times 10^{-5}$) found in the *proteinaceous extracellular matrix* ($p \le 3 \times 10^{-5}$) with molecular functions related to *ion channel binding* ($p \le 1 \times 10^{-3}$) and *collagen binding* ($p \le 2 \times 10^{-3}$). The *p*-values reported for all GO terms are

uncorrected Fisher's exact tests. Looking at the histological images associated with the most extreme positive and negative values of the component, we see dramatic differences in the structure of the stained tissues (Fig. 2a). In particular, images with

high magnitude positive values have well-differentiated nuclei (dark purple spots) and minimal extracellular connective tissue, whereas images with high magnitude negative values have few nuclei and a dramatic presence of extracellular connective tissue (pink colors). Although we do not have access to the pathological image preparation, it appears that these images contain substantial amounts of necrotic tissue. Nonetheless, this component appears to capture differences in the extracellular connective tissue structure, reflected in the extreme-valued histological images and the GO functional terms enriched in the subset of non-zero genes.

The components estimated using supervised ImageCCA that includes an MLP are well correlated with those from unsupervised ImageCCA (Supplementary Figs. 6 and 7). The third ImageCCA-MLP component—which has gene sets corresponding closely to multiple components in the unsupervised ImageCCA results—identifies a set of genes enriched for *blood vessel development* and *vasculature development* that are found in the *extracellular matrix* and are involved in *growth factor binding*. The genes in this component are primarily expressed in testis, EBV-transformed lymphocytes, fibroblasts, and whole blood in the GTEx data. This suggests that the differences in the number or proportion of endothelial and hematopoietic cells—the cells responsible for vasculature development—are captured in this component.

**LGG pathological analysis**. In the LGG pathological image and gene expression data, we extracted 100 CCA components. In the unsupervised setting, these components included an average of 228 genes and 31 image features. The PVE of the components shows fairly monotone decay across the 100 components (Supplementary Fig. 4b). In the supervised setting, these components included an average of 399 genes and 5 image features.

We performed GO term enrichment tests with the subsets of non-zero genes for each LGG unsupervised component. Among the enriched GO terms, we found a diversity of functional categories and cellular localization (Table 1). For example, the enriched terms for the first component of the LGG data are indicative of RNA metabolism (Fig. 2b; Supplementary Tables 3 and 4). In particular, the top component includes genes enriched for *RNA processing* ($p \leq 1.4 \times 10^{-5}$), *lipid particle organization* ($p \leq 2.9 \times 10^{-5}$), and *regulation of DNA metabolic process* ($p \leq 6.7 \times 10^{-5}$). As with the BRCA images, necrotic tissue is visible in the extreme images (Fig. 2b; two smaller images in the positive extreme). These results suggest that this component may reflect technical covariates, such as the time between sample extraction and processing or the proportion of necrotic tissue, where genes involved in RNA decay are correlated with image features that show the morphological effects of time on the tissue sample.

The second component includes genes enriched for *synaptic transmission* ($p \leq 1.3 \times 10^{-23}$), *synaptic signaling* ($p \leq 1.3 \times 10^{-23}$), *trans-synaptic signaling* ($p \leq 1.3 \times 10^{-23}$), and *cell–cell signaling* ($p \leq 5.6 \times 10^{-18}$). This second component includes 77 genes and 38 image features. Many of the genes in this list are only expressed in brain tissues. These clusters can also be used to understand the role of clustered genes without brain-specific function. For example, *SULT4A1* is a sulfotransferase that, in the GTEx data, is primarily expressed in brain samples; furthermore, the Human Protein Atlas shows that the protein is localized to neuronal cells and, specifically, occurs in cytosol[26,27]. While the brain-specific function of *SULT4A1* is unclear, the clustering of this gene with other genes involved in brain synaptic activity suggests that it may be involved in modulating the function of hormones in neuronal cells[27].

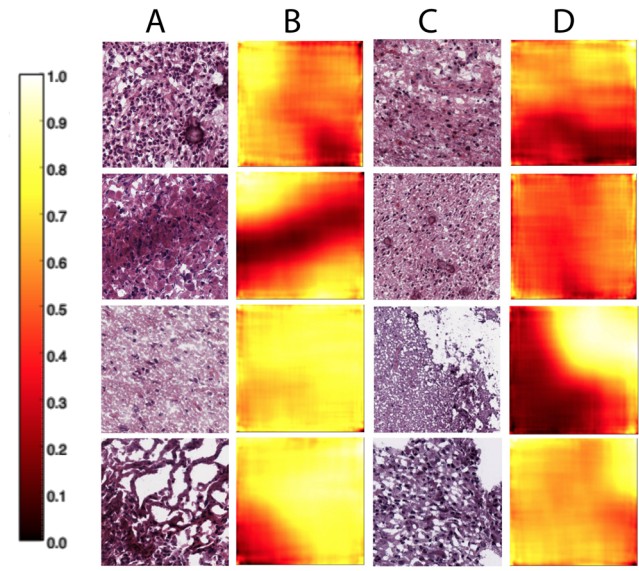

**Fig. 3 Classifying primary versus recurrent tumor locations in LGG histopathological images.** For each overlapping 128 × 128 patch in a 1000 × 1000 pixel image, we classify the likelihood that the patch contains recurrent tumor cells. We used these predictions to create a heatmap of primary tumor (values closer to zero) versus recurrent tumor (values closer to one) locations in the image. Columns A and C are LGG histopathological images; columns B and D are the corresponding heatmaps showing the locations in the images classified as higher likelihood of primary tumor (darker colors) versus higher likelihood of recurrent tumor (lighter colors).

The ninth component includes 97 genes enriched for immune system function: *immune response* ($p \leq 3.9 \times 10^{-29}$), *immune system process* ($p \leq 1.9 \times 10^{-27}$), *defense response* ($p \leq 2 \times 10^{-21}$). The genes in this component are most often expressed in whole blood in GTEx, instead of brain; for example, pleckstrin (*PLEK*) is expressed primarily in whole blood and lymphoblastoid cell lines (LCLs; Supplementary Fig. 8). The extreme-valued images for this component appear to show differences in the proportion of cells from whole blood in the brain tissue section. These results suggest that components are able to identify cell type heterogeneity in brain tissues, and this component specifically captures differences in morphology of brain tissues due to differences in the proportion of whole blood in the sample.

The supervised image feature embedding allows another opportunity for exploratory data analysis with the pathological images. With a trained classifier, we can classify each 128 × 128 patch of an image in terms of whether *primary tumor* tissue or *recurrent tumor* tissue is visible in that patch. Performing this dense classification on the LGG images, we find that supervised ImageCCA is able to annotate the image indicating where in the tumor tissue are primary tumor cells and recurrent tumor cells (Fig. 3). While not the primary goal of our analysis, these results suggest that supervised ImageCCA can be used to segment pathological images with image-level labels that denote what type of cancerous cells are present in an image[28].

**GTEx histological image analysis**. In the GTEx paired histological image and gene expression data, we identified 100 CCA components using ImageCCA. In the unsupervised setting, these components included an average of 1054 genes and 148 image features. The PVE of the components shows non-monotone decay across the 100 components (Supplementary Fig. 4c). We ran the supervised version of ImageCCA on the GTEx data, but found that the accuracy of predicting most tissue labels from

**Table 1 Enriched GO terms for genes selected by sparse CCA in the LGG data.**

| CCA var | GO ID | Term | *p*-value |
|---|---|---|---|
| 1 | GO:0006396 | RNA processing | 1.4e−5 |
| 1 | GO:0034389 | Lipid particle organization | 2.9e−5 |
| 1 | GO:0051052 | Regulation of DNA metabolic process | 6.7e−5 |
| 1 | GO:0051054 | Positive regulation of DNA metabolic processes | 8.6e−5 |
| 2 | GO:0007268 | Synaptic transmission | 1.3e−23 |
| 2 | GO:0099536 | Synaptic signaling | 1.3e−23 |
| 2 | GO:0099537 | Trans-synaptic signaling | 1.3e−23 |
| 2 | GO:0007267 | Cell–cell signaling | 5.6e−18 |
| 3 | GO:0007272 | Ensheathment of neurons | 5.4e−8 |
| 3 | GO:0008366 | Axon ensheathment | 5.4e−8 |
| 3 | GO:0042552 | Myelination | 8e−7 |
| 3 | GO:0032060 | Bleb assembly | 1.4e−5 |
| 4 | GO:0006396 | RNA processing | 6.5e−11 |
| 4 | GO:0090304 | Nucleic acid metabolic process | 5.5e−10 |
| 4 | GO:0034641 | Cellular nitrogen compound metabolic pro… | 7.5e−9 |
| 4 | GO:0006807 | Nitrogen compound metabolic process | 1.1e−8 |
| 5 | GO:0035589 | G-protein coupled purinergic nucleotide … | 2.8e−7 |
| 5 | GO:0035590 | Purinergic nucleotide receptor signaling… | 2e−6 |
| 5 | GO:0035588 | G-protein coupled purinergic receptor si… | 2.4e−6 |
| 5 | GO:0035587 | Purinergic receptor signaling pathway | 8.3e−6 |
| 6 | GO:0044802 | Single-organism membrane organization | 3.6e−6 |
| 6 | GO:0006810 | Transport | 7.3e−6 |
| 6 | GO:1902578 | Single-organism localization | 7.3e−6 |
| 6 | GO:0044765 | Single-organism transport | 7.4e−6 |
| 7 | GO:0006811 | Ion transport | 8.9e−7 |
| 7 | GO:0030029 | Actin filament-based process | 9.3e−7 |
| 7 | GO:0044765 | Single-organism transport | 1.5e−6 |
| 7 | GO:0048771 | Tissue remodeling | 2.5e−6 |
| 8 | GO:0010001 | Glial cell differentiation | 1.6e−5 |
| 8 | GO:0048709 | Oligodendrocyte differentiation | 3.1e−5 |
| 8 | GO:0042063 | Gliogenesis | 8.7e−5 |
| 8 | GO:0042552 | Myelination | 1.2e−4 |
| 9 | GO:0006955 | Immune response | 3.9e−29 |
| 9 | GO:0002376 | Immune system process | 1.9e−27 |
| 9 | GO:0006952 | Defense response | 2e−21 |
| 9 | GO:0002682 | Regulation of immune system process | 1.1e−20 |

Enriched Biological Process GO terms were found separately for each gene set contributing to the first nine CCA components for the LGG data. Only the four most enriched terms per gene set are shown. Uncorrected *p*-values for the Fisher's exact test are reported. Full results shown in Fig. S6.

image features in a test set was almost as poor as random guessing. The challenge of predicting tissue labels from images was noted in prior work[29].

In the CAE used to identify image features in the GTEx data, we examined the convolutional filters to study the patterns identified in the features (Supplementary Fig. 9). The first layer of filters identifies corners and blobs that are indicative of cell nuclei (Supplementary Fig. 9b and d). The second layer identifies various resolutions and spacings of nuclei in a stained image (Supplementary Fig. 9f–h). The third and fourth layers appear to identify different patterns in cell shape and larger contrasting morphological features (Supplementary Fig. 9k, l, p, and q). While these convolutional filters do not allow a precise interpretation of the image features identified by the CAE, they suggest important patterns in specific cellular structures among the histological images.

In the GTEx v6 study data, many of the unsupervised ImageCCA components capture image features and genes specific to a tissue (Supplementary Tables 5 and 6). For example, the first component differentiates skeletal muscle tissue on one extreme from pancreatic tissues on the other via muscle-specific genes (Fig. 2c); the two tissue types have distinct morphology. The genes that are non-zero in this component are highly enriched for *respiratory electron transport chain*, *ATP synthesis coupled electron transport*, and *small molecule metabolic process* (all three

$p \le 2.2 \times 10^{-16}$), *catalytic activity* ($p \le 2.9 \times 10^{-12}$), *oxidoreductase activity* ($p \le 3.2 \times 10^{-10}$), and *endoplasmic reticulum part* ($p \le 2.2 \times 10^{-16}$). We can validate this further by quantifying expression of the genes across the GTEx tissues: the 1630 genes in this component have enriched expression in skeletal muscle (Supplementary Fig. 10). These genes and image features correlate with ischemic time and mode of death (Fig. 4).

The 21st component in the unsupervised ImageCCA distinguishes cerebellum and cerebral cortex tissues from other tissue types—the most extreme tissues are skeletal muscle and pancreas (Fig. 2c). The extreme valued cerebellum and cerebral cortex images include tissues with uniform neurons and densely packed nuclei, while the other extreme is tissues large, long cells (skeletal muscle), or heterogeneous cells (pancreas). The genes in this component are enriched for terms related to synaptic function and localization, including *gated channel activity*, *chemical synaptic transmission*, and *anterograde trans-synaptic signaling*, and *synaptic membrane* (all $p \le 2.2 \times 10^{-16}$). There are 1360 genes associated with this component, and these genes tend to be expressed primarily in cerebellum and cerebral cortex (Supplementary Fig. 10). This component appears to have substantial correlation with ischemic time relative to earlier components (Fig. 4); nonetheless, the associations have a clear biological interpretation outside of ischemic time.

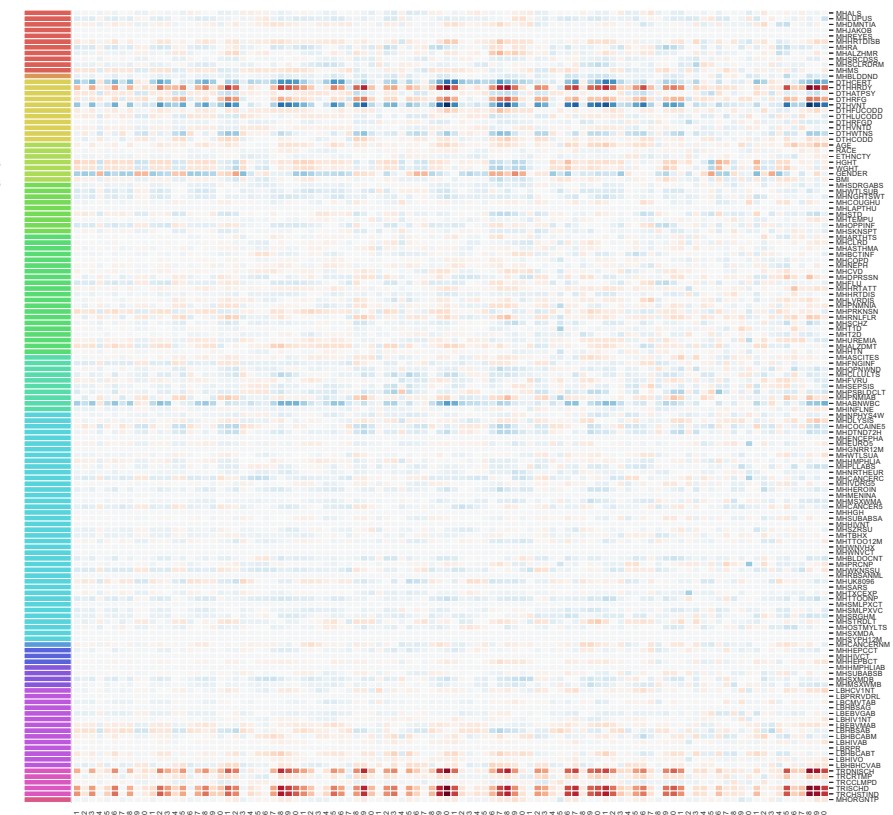

**Fig. 4 Pearson's correlation of 100 components of GTEx CCA with GTEx covariates.** The 100 GTEx CCA components are ordered on the *x*-axis; 138 available GTEx covariates are on the *y*-axis. The legend on the left refers to the Pearson's correlation between each component and the GTEx covariates. Some of the CCA components were sign-flipped so that the Pearson's correlation with covariate Chest Incision Time was non-negative without loss of generality. The colors correspond to covariates in one of the following categories: Autoimmune, Degenerative, Neurological (red), Blood Donation (orange), Death Circumstances (yellow), Demography (yellow green), Evidence of HIV (light green), General Medical History (green), History at Time of Death (sea green), Information (light blue), Medical History (blue), Potential Exposure: Physical Contact (royal blue), Potential Exposure: Sexual Activity (purple), Serology Results (fuchsia), Tissue Recovery (pink), and Tissue Transplant (pink red).

The 24th component in the unsupervised ImageCCA distinguishes testis tissue from other tissues, including muscle tissue (Fig. 2c). The genes in this component are enriched for terms related to spermatogenesis, including *sexual reproduction*, *male gamete generation*, *spermatogenesis*, and *gamete generation* (all $p \leq 2.2 \times 10^{-16}$). There are 1360 genes associated with this component, and these genes tend to be expressed primarily in testis samples (Supplementary Fig. 10). This component appears to have minimal correlation with ischemic time relative to earlier components, but has greater correlation with lupus and type 1 diabetes status relative to the other components (Fig. 4).

Using CCA allows the exploration of components and their relationship to technical and biological factors that confound association tests between single genes and image features. In cis-eQTL mapping—or testing for the association of a genetic variant with expression levels of a nearby gene—the variation in gene expression levels caused by a cis-eQTL is often minimal and local. It is standard in association testing to correct for the principal components (PCs) of the gene expression matrix (or equivalent[30–32]). These PCs are known to capture large effect confounding on the gene expression levels due to batch, platform, cell type composition, donor age, sex, or ancestry. This approach to association mapping controls false positives (spurious cis-eQTLs) while generally not creating false negatives (missed eQTLs).

In our association of image features with gene expression levels, biological signal and technical or biological artifacts may explain a large proportion of the variation in the two observations; thus, controlling naively for principal components of gene expression and image features may remove important shared latent signal. We instead model these confounders jointly with biological signal within CCA, and add sparsity to CCA for interpretability of identified components. For 138 available covariates from the GTEx Consortium, we find substantial correlations with the 100 CCA components (Fig. 4). Many of the components capture variation in features correlated with the same subset of covariates, primarily surrounding type of death, sample ischemic time, and sex, age, and weight. Moreover, despite the sets of genes in each component being mostly different, we note that the correlation signs (i.e., positive or negative correlations) are, for the most part, consistent within the covariates across the CCA components. In other words, across these components, e.g., sex and ischemic time consistently have opposite correlation signs. This consistency implies that there are a large number of gene sets that are involved jointly in the biological processes of postmortem decay and sexual dimorphism with signatures in tissue morphology.

The GTEx v6 study includes genotype data for each donor, allowing us to discover genotypes that are associated with histological image features. To do this, for a specific CCA component, we considered cis-eQTLs in the GTEx data for each gene with a non-zero weight in that component. We tested for association of those cis-eQTLs with the image features with non-zero weights in that component for each tissue. We subset the image samples by tissue and require that each tested SNP be a cis-eQTL in that particular tissue. We require also that the non-zero

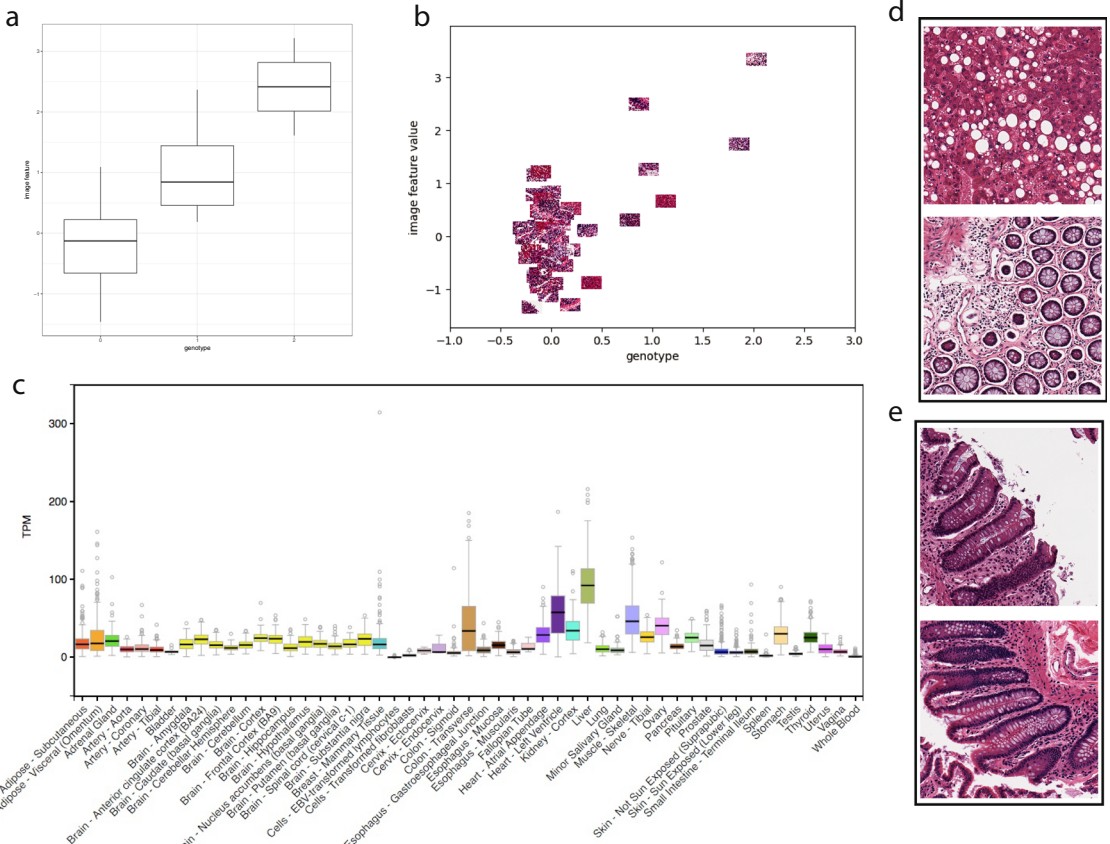

**Fig. 5 Genotype and image feature association for an eQTL targeting lactate dehydrogenase D (*LDHD*) in colon samples. a** Boxplot of association between genotype rs8059637 (*x*-axis) and image feature 799 values for all samples (*y*-axis), the box hinges are the first quartile, the median, and third quartile of the image feature values, respectively, the lower whisker ranges from the bottom hinge to no less than 1.5*IQR (inter-quartile range), the upper whisker ranges from the top hinge to no more than 1.5*IQR; **b** same axes as (**a**), but points are the colon images with jitter added to separate the images; **c** relative abundance of *LDHD* expression across GTEx tissues, with colon—traverse showing substantial expression levels, boxplot defined the same as (**a**) with outlier points defined as greater than or less than the whisker range; **d** images in the top 10% of values for image feature 799; **e** images in the bottom 10% of values for image feature 799.

genes in a given CCA component have non-zero expression in the tested tissue.

For each component, we standardized the values of each non-zero image feature across the *n* images, and we used a linear model to test for association of each feature with the cis-eQTL genotype for the sample donor. Testing for associations within single tissue types reduced the number of available samples to small numbers for many tissues, limiting our power to detect associations. However, we did not find image morphology QTLs that spanned heterogeneous tissue types, so we chose to test within tissues despite power limitations. In particular, we tested for associations in the 15 tissues with at least 20 samples. We calculated false discovery rates (FDRs) of these associations using the Benjamini–Hochberg procedure[33].

Performing association testing in the GTEx data, we found 509 genotype-image feature associations (image morphology QTLs, or imQTLs) including 15 unique mediating gene-image feature pairs in five of the 15 tissue types we tested (FDR ≤ 0.1; Supplementary Table 9). No imQTLs were shared across tissues, and no image features had more than one mediating gene. While the interpretation of the image feature phenotypes in our model with respect to phenotypic differences in tissue morphology is difficult due to the lack of interpretability of the CCA image features, we describe two compelling image morphology QTLs here.

A cis-eQTL for lactate dehydrogenase D (*LDHD*), rs8059637, is associated with image feature 799 (FDR ≤ 0.1) in transverse colon samples (Fig. 5a, b). *LDHD* is an enzyme that converts pyruvate to D-lactate when oxygen is limited during the final step of glycolysis; high levels of lactate reduce the rate of conversion. The Human Protein Atlas shows that the LDHD protein localizes in the cytosol, and is expressed in endothelial and glandular cells in colon samples[34]. Across the GTEx tissues, *LDHD* is expressed in a number of tissues, including liver, skeletal muscle, and transverse colon (Fig. 5c). Visualizing the most extreme positive and negative value colon tissue samples for this image morphology feature (Fig. 5d, e) shows clear differences between the two image extremes. Furthermore, *LDHD* is downregulated in colorectal cancer, and is also a hot spot for somatic mutations in colorectal cancer[35,36].

We also found an association between a cis-eQTL for death-associated protein 3 (*DAP3*), rs4601579, and histological image feature 820 in thyroid tissue (FDR ≤ 0.1; Supplementary Fig. 11a, b). *DAP3* is a mitochondrial ribosomal protein that induces cell death; *DAP3* may be responsible for mitochondrial maintenance rather than protein translation[37]. *DAP3* is expressed across most of the GTEx tissues (Supplementary Fig. 11c). Visual inspection of the images with image feature values at the extremes show differences in cell type composition, nucleation, and extracellular matrix patterns (Supplementary Fig. 11d, e). Previous work

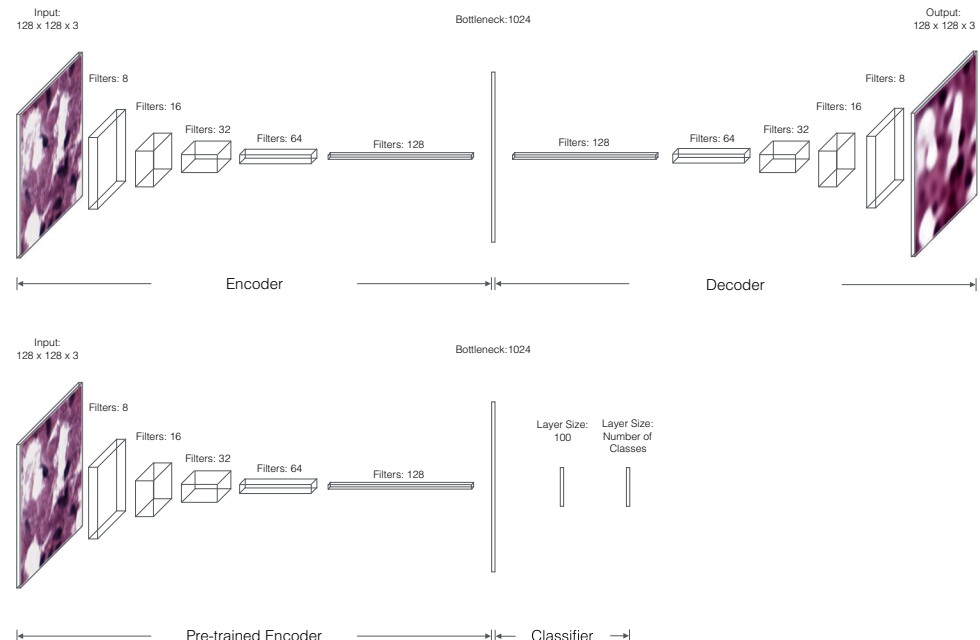

**Fig. 6 Architecture of the CAE.** Each convolutional layer of the encoder includes 5 × 5 filters followed by 2 × 2 max pooling and rectified linear (ReLU) activations. The final convolutional layer of the encoder is fully connected to a layer of 1024 units to produce our embedding. Each convolutional layer in the decoder is upsampled 2× before again applying ReLU nonlinearities. The first convolutional layer of the decoder is linearly projected and reshaped from the bottleneck layer. Bottom: Architecture of the CAE including a multilayer perceptron. The pre-trained encoder is attached to two fully connected layers to allow label classification. The first classification layer features 128 ReLU units, and the second has as many neurons as there are classes with softmax activation (for multi-class problems) or a single sigmoid unit (for binary classification problems).

surveyed the role of *DAP3* in thyroid oncocytoma, a tumor type enriched for mitochondria, in which mitochondrial biogenesis is widespread, and found that *DAP3* was upregulated in tumors undergoing mitochondrial biogenesis, suggesting that *DAP3* may play a role in restricting mitochondrial growth in healthy thyroid cells[37]. Taken together, these results suggest that this imQTL may distinguish thyroid tissues with larger and more numerous mitochondria from those with standard mitochondrial profiles.

## Discussion
In this study, we developed an analysis framework, ImageCCA, for paired histopathological images and gene expression levels to identify subsets of genes that are associated with specific features of tissue morphology. We applied this framework to three sets of paired histological image and gene expression data: breast carcinoma samples, LGG samples, and GTEx v6 tissue samples. The choice of the method to engineer histological image features that were concise and meaningful with respect to genomic data was crucial both for finding associations and for biological interpretation of those results. Applying the ImageCCA framework to these data, and interpreting the components, we were able to find genes known to influence cellular morphology, including the extracellular matrix and the cell wall, and involved in tissue-specific morphology, including neuronal, testis, and muscle tissue.

On tumor sample data, we used ImageCCA with a multi-layer perceptron to segment the pathology image in order to highlight the image locations with features predictive of tumor sample morphology. We validated our ImageCCA findings using GO term enrichment analyses and correlations with held-out sample data. Our results demonstrate that biologically meaningful correlations exist and can be identified between gene expression and features extracted from histological images. It is still uncertain whether the products of these genes are directly or even indirectly responsible for the visible features, or whether they are jointly influenced by a shared latent component, such as ischemic time

or exposure. Additional analyses may be used to identify a causal effect of the gene expression levels on image morphology, such as Mendelian randomization techniques with known eQTLs[38].

We have shown that the framework introduced here can be applied to both pathological and healthy tissue samples, and to both single tissue types and a mixture of types, to detect correlations between gene expression and image features. We note that we identify correlations here and do not make causal statements about the relationship between gene expression and cellular morphology; exciting experiments that modify cell shape find changes in gene expression levels[39]. A number of observations—methylation levels or cis-regulatory element information, for example—could be included in these analyses by using methods for group factor analysis, which allow more than two sets of paired observations to be included in the same type of sparse CCA[21,25].

While we have tried to interpret the image features extracted from the CAE, the interpretations are not straightforward. One option is to extend the image feature extraction process to include features identified by tools designed for quantitative analyses of these histology images[40]. A second caveat is that the supervised labels (tissue label; cancer status) do not capture the image features that are associated with gene expression and genotype. Instead, we would ideally use the high-dimensional gene expression and genotype values themselves to supervise the automatic extraction of image features, ensuring that the image features will represent characteristics of the images that are best correlated with the high-dimensional genomic observations[29,41,42].

The connection between variation in gene expression levels and in the corresponding tissue image suggests that one can be used to aid in the analysis and prediction of the other. A pathologist who visually inspects tissue images for diagnostic purposes could confirm each observation using predicted expression values of the genes linked to the visible feature of interest. Conversely, in some cases clinically significant values in a patient's gene expression

profile could be used to generate an encoding of the visual properties of the associated histological image. This study begins to address the question of how regulation of gene expression in tissues relates to tissue morphology and downstream organismal phenotypes.

## Methods

**Processing the image data**. Image processing was performed using the ImageJ software package[43]. The images, which were in SVS format, were imported using the Bio-Formats plugin[44], and split into $1000 \times 1000$ pixel tiles. Raw images used inconsistent magnification, but in general images were taken with either ×20 apparent magnification, resulting in around 0.25 microns per pixel, or ×40 apparent magnification, resulting in around 0.50 microns per pixel.

A $1000 \times 1000$ tile was considered for selection if the mean gray values of itself and the tiles above, below, left, and right of it were each below (darker than) 180 out of 255. Because of the intractable file size of the full-resolution images, tile selection was actually performed on the ×16 lower resolution version of the image, and the region in the full-resolution image corresponding to the selected tile was extracted.

At each layer in the encoding section of the CAE, convolution followed by max pooling results in halving incoming data in the length and width dimensions. We double the number of convolutional filters at each successive layer (Fig. 6). The decoding section of the CAE mirrors the encoding section. To train the autoencoder, we fed randomly cropped and rotated $128 \times 128$ windows of the processed images into the network and trained it to minimize the mean squared error between its output and input. The advantage of this sampling procedure is two-fold; we are able to overcome the challenges of reconstructing large images while also synthetically increasing the size of our training data.

Once the network was trained, each image was represented by randomly sampling a hundred $128 \times 128$ windows from it, embedding each using the encoding section of the CAE, and averaging those feature encodings (for $m = 100$, $i \in \{1, ..., n\}$ samples, and $j \in \{1, ..., p\}$ image features):

$$y_{i,j} = \frac{1}{m} \sum_{\ell=1}^{m} x_{i,j,\ell}, \qquad (1)$$

where $y_{i,j}$ is the average learned representation for sample $i$ and dimension $j$, and $x_{i,j,\ell}$ is the $\ell$th sampling of the representation for sample $i$ and dimension $j$. Because the CAE is trained to reconstruct images as accurately as possible, some variance of the encoded samples are inevitably used to represent the locations of structures in image, while the remainder is used to represent the physical properties of those structures. This averaged bag of features representation allows us to essentially integrate away the location-based variance, while keeping information about the image properties in which we are most interested. The 1024 feature vector was the mean encoded feature value across the 100 image windows of each image. Finally, we whitened the averaged image representations using PCA[45]. We use the 1024 whitened features to represent the images in CCA (Supplementary Fig. 1). This procedure decorrelates each dimension of the feature space, which is helpful for interpreting the results of CCA.

**Supervised feature extraction using multi-layer perceptron**. In supervised ImageCCA, we added a MLP to the encoding pipeline of the CAE. We trained the MLP to identify features that distinguish histological images according to the image labels. In particular, we used the activations at the last hidden layer of the aggregate network as the supervised image features in the downstream CCA in place of the image features from the unsupervised CAE. As with the CAE, we took the average of the 100 windows of the MLP, and we whitened the supervised image features using principal components analysis.

**Cancer and tissue classification**. To capture variance that corresponds to the presence of cancer (BRCA, LGG) or to the tissue type (GTEx), we simply created a new network consisting of the pre-trained encoding section of the CAE and two fully connected layers for classification (Fig. 6). After retraining this network to minimize the log loss between the predicted label and the true label, the encoding module learns to represent images in terms of features useful for cancer or tissue classification rather than image reconstruction.

We then performed the same averaging and whitening steps with the newly trained encoder in order to obtain a final image representation.

**Gene expression level preparation**. The RNA-Seq sample libraries had been prepared using the Illumina TruSeq Kit and paired-end sequencing was performed on the Illumina HiSeq2000. For the BRCA and LGG data sets, the RSEM algorithm was used within the SeqWare framework to estimate the fraction of transcripts in the sample belonging to each gene[46], while RPKM values were calculated for the GTEx data set. These values were log transformed and then scaled such that the values for each gene have mean zero and standard deviation one so that the CCA coefficients for these variables are comparable. Genes with zero variance were removed.

**Sparse CCA**. A consideration in the selection of a method for high-dimensional correlation analyses was the ability to capture, partition, and control variation in the two sets of observations, and then use downstream analyses of the results to interpret the source and type of the variation. To do this, we applied sparse CCA to the extracted image features and paired gene expression values as the two sets of observations. For the supervised setting, CCA was performed with the same gene expression values, but with the classification-transformed image representations, using the same parameters. CCA is a linear projection of two sets of observations into a shared latent subspace that maximizes correlation between the sets[20,47].

For results reported here, we used the SPC implementation of sparse CCA[48] using the CCA function in the PMA R package[49] with gene expression values and image representations as the two sets of variables to be correlated (Supplementary Figs. 2 and 3). In this framework, CCA components are iteratively identified conditional on the previous components, which encourages uncorrelated components that explain sequentially and stochastically less variation in the original observations (Supplementary Fig. 4). We fix the number of components $K = 100$ for all three data sets because (i) this method is greedy and deterministic, so we can choose the appropriate $K$ up to 100 without affecting results for $K < 100$; (ii) we see interesting biological signal in the later components; (iii) we observed heterogeneous levels of sparsity across the components, capturing different classes of variation; and (iv) because proportion of variance explained (PVE) decays in a non-monotone way, thresholds on PVE or similar metrics are not meaningful.

**Hyperparameter tuning**. Sparse CCA requires setting three hyperparameters: $\lambda_1$, and $\lambda_2$, the amount of sparsity regularization applied to the image feature and gene expression matrix, respectively, and $K$, the number of CCA components. To select values, we performed a hyperparameter search for both $\lambda$ values (Supplementary Fig. 2) in the LGG data set. We evaluated the quality of the parameter settings using Pearson's correlation between the image reconstructed using CCA and the true image, and between the gene expression levels reconstructed using CCA and the true gene expression levels. These results imply that greater sparsity is more important when predicting genes, likely because there are many more features in this space.

In this work, we are primarily interested in selecting sparsity parameters that allow optimal reconstructions of images and gene expression levels, that produce a small number of genes and image features per component, and that produce interpretable subsets of genes as quantified by GO term enrichment. Using a grid search, we fix $\lambda_1$ (for image features) to 0.15 for all applications and $\lambda_2$ (for gene expression) to 0.05 for BRCA and LGG and to 0.10 for GTEx (Supplementary Fig. 2). We validated the robustness to selection of these hyperparameters by looking at correlations among the components from two different hyperparameter settings (Supplementary Fig. 3).

**Gene set enrichment analysis**. The gene sets selected by sparse CCA were tested for enriched GO[50] terms using the topGO[51], org.Hs.eg.db[52], and GO.db[53] R packages.

**Tissue-specificity of gene expression**. We investigate the gene-tissue specificity of the top 100 CCA components by plotting a heat map of the normalized gene expression level across each of the tissues with images in the GTEx dataset. For a given CCA component, we consider only the genes with a non-zero loading in that component. For each gene, we compute an average gene expression value in that tissue by averaging the gene expression level across all samples present in that tissue. We normalize the average gene expression value for each gene across all tissues by the $\ell_1$ norm of that gene, such that the values for each gene across all tissues sum to one, and the maximum value per gene is one. For comparison purposes we append another column of expression values for whole blood despite not having any image samples of whole blood. A tissue with high normalized gene expression values across all the genes in a given CCA component implies that the component is largely tissue-specific.

**GTEx histological image association mapping**. To identify genotype–image feature associations between the GTEx genotype data and GTEx histological images, we perform association mapping between the SNPs and image features relevant to the top CCA components. Each latent component (in gene expression space or image feature space) that the sparse CCA method aims to estimate is a linear combination of the original expression or image features; thus, for each CCA component, there are weights on the original features that are either zero or non-zero. For downstream association analysis, we consider only the top 100 CCA components with the strongest correlations, and within those components only the genes and image features with non-zero weights. For association mapping, we use the 1,289,112 SNPs that are a known cis-eQTL for at least one of the 18,204 genes in the top 100 CCA components with a non-zero weight, and we use 926 out of 1024 image features in the top 100 CCA components with a non-zero weight. We normalize the image features by projecting each feature onto the quantiles of the empirical distribution over all features.

Association mapping is conducted by performing all pairwise univariate linear regressions between those remaining SNPs and image features, using MatrixEQTL[54]. Since each image is only of one tissue, we test for putative

associations only between those SNPs that are previously identified cis-eQTL for a gene in the same tissue that corresponds to the sample image. For each association test, we compared the null hypothesis of no association ($\beta = 0$) versus a non-zero association between genotype and image feature ($\beta \neq 0$), and calculated the p-value of the T-statistic corresponding to the value of $\beta$ and the standard error of the regression. From these association statistics, the false discovery rate (FDR) was computed via the Benjamini–Hochberg procedure[33] using the `multipletests` function in the `statsmodels` Python package[55]. We chose to threshold at FDR < 0.1 based on prior work[9], meaning that 10% of the discoveries will be false positives in expectation.

**Reporting summary**. Further information on research design is available in the Nature Research Reporting Summary linked to this article.

## Data availability

Tissue slide images for the BRCA and LGG samples were downloaded from the TCGA Data Portal (now available at the GDC Legacy Archive: https://portal.gdc.cancer.gov/legacy-archive/). Details of the data collection and preparation can be found in the original studies[22,23]. Tissue slide images for the GTEx samples were downloaded from the NCI Biospecimen Research Database (https://brd.nci.nih.gov/brd/image-search/searchhome). Genotype and RNA-Seq data are available upon application from dbGAP (https://www.ncbi.nlm.nih.gov/projects/gap/cgi-bin/study.cgi?study_id=phs000424.v7.p2).

## Code availability

The software for the novel analysis in this manuscript is publicly available at: https://github.com/daniel-munro/imageCCA. We used the following Python packages with corresponding version numbers: `statsmodels` (0.9.0). We used the following R packages with corresponding version numbers: `topGO` (2.30.1), `Bioconductor` (3.6), `org.Hs.eg.db` (3.5.0), `GO.db` (3.5.0), `PMA` (1.0.9), and `MatrixEQTL` (2.1.0).

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

## Acknowledgements

B.E.E. was supported by NIH R01 HL133218, a Sloan Faculty Fellowship, NSF CAREER AWD1005627, CZI AWD1005664, and CZI AWD1005667. D.M. was funded by the NSF Graduate Research Fellowship Program under Grant No. DGE 1148900. The input data reported in this paper are archived at The Cancer Genome Atlas (TGCA), study accession IDs TCGA-BRCA (BRCA) and TCGA-LGG (LGG), and dbGaP phs000424.v6 (GTEx). We acknowledge the kind help of Phil Branton, the pathologist for the GTEx project.

## Author contributions

J.T.A., D.M., and B.E.E. conceived the experiments. J.T.A., D.M., G.D., and B.E.E. designed the experiments. J.T.A., D.M., and G.D. performed the computational experiments. D.M., G.D., and B.E.E. analyzed the results. All authors wrote the paper. G.D. and B.E.E. revised the paper.

## Competing interests

B.E.E. is on the SAB of Freenome, Celsius Therapeutics, and Creyon Bio, and is a consultant for Genomics plc and Freenome. The remaining authors have no competing interests to declare.
