## [Peer Review File · Nature Communications]

Reviewer #1 (Remarks to the Author):

Summary of paper

The paper introduces ImageCCA, a framework that jointly analyses paired gene-expression levels and histological/histopathological image phenotypes to identify subsets of genes that describe tissue phenotypes.

The method uses a) convolutional auto encoders (CAE) that extracts image features per input image, by transforming the image to subsequent lower-dimensional spaces (encoder) and then reverse-transforming the image to higher-dimensions (decoder) for image reconstruction and b) Sparse canonical correlation analysis (CCA) that gives a linear projection of gene subsets and image feature subsets onto a shared latent subspace that maximises correlations between these subsets.

The method is demonstrated on GTEx and TCGA datasets where interpretable gene subsets are extracted along with meaningful image feature subsets.

The paper provides for a good read and is a neat application of CAE and Sparse CCA to paired gene-expression and imaging datasets.

Comments

1. Why is K (#CCA components) fixed at 100 for all the datasets especially when the three datasets have different number of images? (The number of genes can be considered similar across the different datasets). Have you chosen this based on model selection/AIC/BIC scores on some goodness of fit criteria?

How do the method's results differ for a larger or smaller K ?

2. How sensitive is your method to the sparseness parameter? How did you determine the current sparseness parameters

3. Given that you use the (sparse) CCA, you are also jointly modelling the technical noise and biological signal present in both data modalities. Other than the results you have demonstrated, how do you justify that the model teases apart the technical noise from the biological signal? Or in other words, that you have not overfit noise into the model, given that CCA is a special form of the multiple regression. Was any regularisation used?

4. Given the supplementary Fig 1, how do we arrive that the bottleneck layer has a dimension of 1024?

5. The description on *XXYLT1*: the last line of Page 15 is unclear. Are you mentioning 'any protein' or 'any Notch protein'.

6. Fig 4 - Why do you take the 'absolute' values of the Pearson's correlation - you would get relationships that are highly 'correlated' or 'anticorrelated' between CCA components and GTEx covariates but by removing the signage, are not these relationships thrown away?

7. FDR - Depending on the FDR, you will get different gene-image feature subsets. How do you decide which value of FDR to pick for a particular dataset?

8. While the application is neat, I am not completely convinced in the "biological interpretation" presented.

Minor comments

1. Decide between 'observation' and 'samples'.
2. Page 13 - First line - Fig 3 instead of Fig 1.
3. Similarly, in the next line, you may want to add 'L4' as well.
4. Page 13, 2nd para, 4th line - Fig 2c instead of Fig 2a.
5. Table S1: what is the difference between 'annotated' and 'selected' columns (referring to the caption).

Reviewer #2 (Remarks to the Author):

The earliest descriptions of cancer involved characterizing gross differences in tissue morphology between cancerous and healthy tissue. Almost immediately after invention of the microscope, differences in tissue and cell morphology became the basis for the field of cancer pathology. Even today, visual examination tumor sections are used as a routine part of diagnosis. With the development of advanced computer vision and machine learning methods, it is becoming increasingly possible to perform automated, and unbiased, cancer diagnoses through imaging of tumor biopsy sections.

However, one outstanding question is: Why does morphology predict disease outcomes? Or more precisely: What are the differences in gene expression and/or signaling network activity that both drive disease and must correlate with changes in tissue and cell morphology? Answering this question is essential because it will allow the identification of disease driving genes/proteins in individual patients that can be targeted therapeutically. An important point regarding these drivers of interest, is that it is highly likely they have already been identified by expression studies. The issue is finding which of these genes have expression changes that correlate with morphological changes.

Although making the link between changes in gene expression and tissue/cell morphology was unfeasible even recently, data sets that have now been generated which have histopathological images and gene expression profiles from the same tumor samples, which now facilitate the establishment of gene-morphology correlations. That said, these data sets are relatively sparse given the high dimensional space that both gene expression and morphology can exist in. Thus, great care must be taken in identifying these correlations.

Towards establishing correlations between expression and morphology, Engelhardt and colleagues describe a methodology using convolutional autoencoders (CAEs) and sparse correlation analysis to identify genes that correlate with subsets of visual features. They deploy this method on three different data sets.

I very much appreciate what these authors are trying to do here, and the challenges this involves. However, in its current form I do not think this manuscript warrants publication in a journal such as Nature Communications with a very broad audience.

Major points:

1) The results

The approach is very logical and importantly I think is likely providing robust results. The authors should be applauded for dealing with two challenges in this type of data. By not attempting to use a conventional segmentation followed by feature derivation approach to analyse images, this allows them to rapidly and relatively accurately process and classify pathology images. A task which would otherwise present a significant technical difficulty. Moreover, the authors are clearly aware of the sparseness of the data, and their method is careful to avoid correlations that could be driven by noise.

However, I think the robustness of the methodology might be coming at a price. A challenge in these type of analyses is finding the real, but sometimes weak, signal amongst the noise. It is in finding these weaker signals that new biological insight lies, and the ability to identify them separates the truly innovative methods apart from the those that may simply be intuitive. Here, I believe the authors are only identifying very (very) strong signals, which do not provide a great deal of biological insight - novel or otherwise. In the case of the BRCA dataset, they identify changes in adhesion and ECM genes; which while expected, are really the only classes they find enriched. In the LGG analysis, the authors find terms associated with RNA metabolism which the authors attribute to sample processing (I am not entirely convinced by their argument that this is the case). Finally, in the GTEX analysis the GO terms are largely associated with muscle differentiation, which correlates with the obvious large differences between normal muscle and tumor morphology. While it is encouraging these obvious differences are being identified, the fact their methodology cannot provide a deeper level of insight makes me question its utility.

Moreover, any validation; of these associations is entirely descriptive in nature, and I am not even 100% sure the authors have taken the care to make sure these associations do in fact make sense. As a case in point, in the BRCA dataset the top component was enriched for genes involved in cell adhesion, proteinaceous extracellular matrix, collagen binding, which qualitatively correlates with the very different changes in tissue architecture between tumor and normal. OK, on the surface this seems promising (although again, completely expected), but do changes in the individual genes and/or classes make biological sense? It is loss of cell-cell adhesion that correlates with disease? Or is it gain of cell-ECM adhesion? Looking at ECM-associated genes becomes much more complicated. Cancerous tissue does not, as the authors seem to imply, arise solely through a loss of normal ECM, but rather a complete reorganization of ECM that is accomplished through changes in ECM proteins, proteases, and ECM-binding proteins. So do the changes in ECM related genes actually reflect this? The authors should also be very careful that a handful of genes is not driving multiple category enrichments.

Answering these questions is important because I think it validates the method, but also may provide some mechanistic insight which is desperately needed here.

Although I do not expect the authors to do everything here, it would be nice if they could show that they can identify genes or targets in a way that I would believe this method would have utility in the future both in fundamental research or in the clinic.

I would suggest the authors try digging deeper into the results of their analyses, and in particular the results of the BRCA dataset, as I believe that is where they are most likely to find something insightful. As a suggestion they may consider using methods alternative to GO enrichment to identify genes/proteins and then look at their output on a gene-by-gene basis.

2) Normal versus graded cancers

Are the authors able to identify only the strongest signals due to the fact that they have included so much normal tissue, especially in the LPP and GTEX analyses? And are often identifying classifying normal versus cancer tissue, instead of low grade vs. high grade?

3) QTL analysis

I do find the attempt to identify associated QTLs VERY interesting, but very preliminary. Unlike some of the results of the correlated gene expression analysis, it is completely if those genes have any meaningful role to play in cancer biology.

4) Other

For a general journal such as Nature Communications, I found this manuscript not easy to go

through (and I am in this area). To make it accessible to a broad audience. authors should spend considerable effort changing the figures and text.

Some notable phrases in particular:

"Canonical correlation analysis (CCA) finds linear mappings for two sets of variables observed in paired samples onto a shared latent subspace upon which they are maximally correlated with each other."

"We interpret the latent semantics of the CCA components by examining the enriched molecular functions and tissue specificity of the subsets of genes and also the differences in cellular morphology of the extreme values of the images."

"We then used the activations at the last hidden layer of the aggregate network as the supervised image features in the downstream CCA in place of the image features from the unsupervised CAE."

Reviewer #3 (Remarks to the Author):

The authors propose and test the combination of two machine learning approaches for inferring gene expression patterns that are associated with histology images. First, using CAE, they capture high-dimensional imaging features, and subsequently they use CCA to jointly infer latent variables from imaging features and gene expression data that co-vary. They apply this approach in varied setting, including data from TCGA and GTEx. They show the existence of expression latent variables that are correlated with histology features.

Overall the method presented combines several interesting approaches with desirable properties, including automatic feature generation from histology images, and accounting for broad patterns that may be related to confounding. However, I have two primary concerns: a) I did not get a good sense of biological usefulness/significance of the method, as in the results were not placed in the context of previous knowledge and finding, b) the statistical significance of the specific correlations reported is not assessed (see below). Please see below for my comments:

- The authors report some imaging based patterns that correlate with expression latent variables, however, there is no discussion of what new biology has been learned (besides the generic GO enrichment analysis) and how it compares with what has been known.
- The organization of the articles makes it hard to read: the Results section contains a mixture of materials, methods, results and discussion, but all separated based on dataset. I suggest the authors re-organize the paper's content.
- Although there is an overall permutation analysis for assessing the overall significance of features, can the author show that each latent variable is statistically significant? As the authors know, with the highdimensionality of the data, sCCA will essentially likely find some shared latent dimensions, but which ones are significant?
- How robust is the method with respect to parameter setting for sCCA?
- GO enrichment analysis some times include terms with pretty large pvalues (e.g., $\sim 10^{-4}$), how many GO terms were analyzed and what's the significance threshold?
- The statistical significance threshold for cis-eQTL analysis (FDR 0.1 and FDR 0.2) is pretty large, please justify.

Response to reviewers

Thank you for these thoughtful and thorough reviews for this manuscript. We have addressed each point raised by the reviewers; reviewers' comments are in plain text and *our responses are in blue italic text*. While we originally were tracking changes to the text with markers, we note that many sections were rewritten so substantially that we decided it was easier to submit a clean document.

Reviewer #1

The paper introduces ImageCCA, a framework that jointly analyses paired gene-expression levels and histological/histopathological image phenotypes to identify subsets of genes that describe tissue phenotypes. The method uses a) convolutional auto encoders (CAE) that extracts image features per input image, by transforming the image to subsequent lower-dimensional spaces (encoder) and then reverse-transforming the image to higher-dimensions (decoder) for image reconstruction and b) Sparse canonical correlation analysis (CCA) that gives a linear projection of gene subsets and image feature subsets onto a shared latent subspace that maximises correlations between these subsets. The method is demonstrated on GTEx and TCGA datasets where interpretable gene subsets are extracted along with meaningful image feature subsets.

The paper provides for a good read and is a neat application of CAE and Sparse CCA to paired gene-expression and imaging datasets.

We thank the Reviewer for the clear and enthusiastic description of our manuscript.

1. Why is K (#CCA components) fixed at 100 for all the datasets especially when the three datasets have different number of images? (The number of genes can be considered similar across the different datasets). Have you chosen this based on model selection/AIC/BIC scores on some goodness of fit criteria? How do the method's results differ for a larger or smaller K ?

This is a good point. The sparse CCA model that we used is a greedy method [1], so, if we used $k < 100$ components in one application of the method, those first k components of the 100 component application will be identical to the k components of the k application. Multiple reviewers pointed this out, and we have clarified this point at multiple locations in the text. We also note that our image morphology QTL in thyroid was identified from component 57, highlighting the biological signal in the second half of the component set.

More broadly, the "right" number of components is a bit difficult to assess, and many latent factor models have tried, but there is no current community standard [?]. The only place we consider all $K = 100$ components is in the bulk statistics of mean and variance in the number of non-zero genes and image features across components. Other than that, we restrict our analysis to the interpretation of the genes and image features that are members of specific components, prioritizing components that explain larger proportions of variance. To be clear though, we found very

interesting components explaining smaller proportions of the variance with correlations to relevant covariates, and we believe that a simple statistical test will not take these downstream analyses into account. To address this issue in a more quantitative way, we have explicitly computed and show in a new figure the proportion of variance explained (PVE) from each component, and a histogram of the genes and image features across the different studies.

A more subtle point, though, that we emphasize here and in the manuscript is that, as the PVE decreases (in expectation, not in a monotone way) and the component numbers increase, the signals may be just as biologically meaningful and interpretable, but confined to the correlated variation of a very small number of genes and image features, and thus only account for a small PVE. We have more thoroughly examined this possibility in the Results.

2. How sensitive is your method to the sparseness parameter? How did you determine the current sparseness parameters?

We performed the unsupervised equivalent of cross-validation to determine the current sparseness parameters, and we have added the details of this analysis to the Results, the Methods, and in Supplementary Figure 2. We found that our approach was not particularly sensitive to the sparseness parameter, and we quantify this robustness in the Supplementary Figure 3. We note that prediction and interpretation are often (and in our application in particular) at odds with each other. Specifically, when we select the sparsity parameter such that the cross-validation is optimal in a grid search, we note that the GO terms identified by the subset of genes have p-values in a range that does not allow a clear interpretation of the collective role of those genes. Nonetheless, we updated our Results to reflect the new runs of ImageCCA in light of the different sparsity parameters selected through cross validation and GO analysis.

3. Given that you use the (sparse) CCA, you are also jointly modelling the technical noise and biological signal present in both data modalities. Other than the results you have demonstrated, how do you justify that the model teases apart the technical noise from the biological signal? Or in other words, that you have not overfit noise into the model, given that CCA is a special form of the multiple regression. Was any regularisation used?

Yes, you are correct that this claim needs further support. We have updated our manuscript dramatically with more thorough analyses, including a clear description of a set of components that are identified that capture ischemic time and mode of death, and another set that reflect variation in cell type composition. Sparsity is a form of regularization, and one that allows the non-zero components to effectively allow the component itself to be interpreted in terms of what type of variation it captures (e.g., variation due to proportions of whole blood cells in the sample or ischemic time of the samples). The more subtle point here is that we fit all variation, regardless of that variation being due to “signal” or “noise” and then interpret the components to understand what source of variation is captured in that component marginally. Yes, regularization was used in the sparsity penalty for CCA.

4. Given the supplementary Fig 1, how do we arrive that the bottleneck layer has a dimension of 1024?

We are not entirely sure what this question means. In the Supplementary Figure 1, the bottleneck layer is represented by the vector in the image, and it is of dimension 1024. If the question is about how we chose this dimension, we looked at current state-of-the-art convolutional autoencoders for image data, and used the bottleneck size from those. We have added this justification in the Results, noting that we also perform whitening of this feature space before performing canonical correlation analysis.

5. The description on XXYLT1: the last line of Page 15 is unclear. Are you mentioning 'any protein' or 'any Notch protein'.

We have removed this statement.

6. Fig 4 - Why do you take the 'absolute' values of the Pearson's correlation - you would get relationships that are highly 'correlated' or 'anticorrelated' between CCA components and GTEx covariates but by removing the signage, are not these relationships thrown away?

Because the CCA components are identifiable up to sign changes (i.e., the sign of the loadings can all be swapped and the sign of each factors can all be swapped and, under the symmetric penalty and likelihood model, the value of the objective function will be identical), the sign of this correlation is meaningless. We have added this clarification in the Fig. 4 caption.

7. FDR - Depending on the FDR, you will get different gene-image feature subsets. How do you decide which value of FDR to pick for a particular dataset?

We are not entirely sure which FDR threshold you refer to here, but the most plausible one is the FDR of image QTL association. For this FDR, we have updated our analyses to chose a threshold of $FDR < 0.1$, and this was applied to all tissues uniformly and only in one data set. We did not choose an FDR to identify gene-image feature subsets; we identified these subsets based on non-zero values in each CCA component. We have clarified this in the text.

8. While the application is neat, I am not completely convinced in the "biological interpretation" presented.

We understand. We have spent the last few months revising the manuscript and working with Phil Branton, the pathologist for the GTEx project who collected these GTEx images. We also removed most of the biological interpretations to avoid overinterpreting our results. We hope that the biological interpretation presented in the revised manuscript is more convincing now.

Minor comments

1. Decide between 'observation' and 'samples'.

Throughout, 'observation' refers to either the set of image features or the gene expression features (e.g., data modality), and 'samples' refers to the single, independent and identically distributed set of values from both types of data modalities. We have reviewed the use of each word in the

manuscript to confirm.

2. Page 13 - First line - Fig 3 instead of Fig 1.

Thank you, we have fixed this typo.

3. Similarly, in the next line, you may want to add 'L4' as well.

Thank you, we have added this reference.

4. Page 13, 2nd para, 4th line - Fig 2c instead of Fig 2a.

Thank you, we have fixed this reference.

5. Table S1: what is the difference between 'annotated' and 'selected' columns (referring to the caption).

Good question. We have updated the caption to make this notation clear.

Reviewer #2

The earliest descriptions of cancer involved characterizing gross differences in tissue morphology between cancerous and healthy tissue. Almost immediately after invention of the microscope, differences in tissue and cell morphology became the basis for the field of cancer pathology. Even today, visual examination tumor sections are used as a routine part of diagnosis. With the development of advanced computer vision and machine learning methods, it is becoming increasingly possible to perform automated, and unbiased, cancer diagnoses through imaging of tumor biopsy sections.

However, one outstanding question is: Why does morphology predict disease outcomes? Or more precisely: What are the differences in gene expression and/or signaling network activity that both drive disease and must correlate with changes in tissue and cell morphology? Answering this question is essential because it will allow the identification of disease driving genes/proteins in individual patients that can be targeted therapeutically. An important point regarding these drivers of interest, is that it is highly likely they have already been identified by expression studies. The issue is finding which of these genes have expression changes that correlate with morphological changes.

Although making the link between changes in gene expression and tissue/cell morphology was unfeasible even recently, data sets that have now been generated which have histopathological images and gene expression profiles from the same tumor samples, which now facilitate the establishment of gene-morphology correlations. That said, these data sets are relatively sparse given the high dimensional space that both gene expression and morphology can exist in. Thus,

great care must be taken in identifying these correlations.

Towards establishing correlations between expression and morphology, Engelhardt and colleagues describe a methodology using convolutional autoencoders (CAEs) and sparse correlation analysis to identify genes that correlate with subsets of visual features. They deploy this method on three different data sets.

I very much appreciate what these authors are trying to do here, and the challenges this involves. However, in its current form I do not think this manuscript warrants publication in a journal such as Nature Communications with a very broad audience.

We are extremely impressed at the thought and care that the Reviewer has put into a description of the background and motivation of the problems described in this manuscript. We hope that our responses and sweeping revisions to the manuscript make it more appropriate for the general audience.

1) The results. The approach is very logical and importantly I think is likely providing robust results. The authors should be applauded for dealing with two challenges in this type of data. By not attempting to use a conventional segmentation followed by feature derivation approach to analyse images, this allows them to rapidly and relatively accurately process and classify pathology images. A task which would otherwise present a significant technical difficulty. Moreover, the authors are clearly aware of the sparseness of the data, and their method is careful to avoid correlations that could be driven by noise.

However, I think the robustness of the methodology might be coming at a price. A challenge in these type of analyses is finding the real, but sometimes weak, signal amongst the noise. It is in finding these weaker signals that new biological insight lies, and the ability to identify them separates the truly innovative methods apart from the those that may simply be intuitive. Here, I believe the authors are only identifying very (very) strong signals, which do not provide a great deal of biological insight - novel or otherwise. In the case of the BRCA dataset, they identify changes in adhesion and ECM genes; which while expected, are really the only classes they find enriched. In the LGG analysis, the authors find terms associated with RNA metabolism which the authors attribute to sample processing (I am not entirely convinced by their argument that this is the case). Finally, in the GTEX analysis the GO terms are largely associated with muscle differentiation, which correlates with the obvious large differences between normal muscle and tumor morphology. While it is encouraging these obvious differences are being identified, the fact their methodology cannot provide a deeper level of insight makes me question its utility.

The Reviewer makes a powerful point: we had only analyzed the components with the largest proportion of variance explained in the data. This, as the Reviewer notes, means that much of the interesting signal, which actually contributes to a small proportion of the variance of the observations, is not identified. In this revised version, we go into much deeper detail to consider the biological implications of each of the components from each of the studies. In particular, in response to these comments, we have included the following new downstream analyses: i) clear description of the proportion of variance explained for each application of ImageCCA (Supplemen-

tary Figure 4); ii) correlation of components within an application of Image CCA (Supplementary Figure 12); iii) correlation of components with all observed covariates within an application of Image CCA (Fig 4); iv) gene set enrichment analyses for each component (Supplemental Data); v) quantifying the tissue specificity of the genes within an Image CCA component (Supplementary Fig 2); vi) image QTL discovery (Fig 5; Supplementary Fig 11). We believe these analyses of the patterns identified in these data better capture the extent of the variation identified in the sparse components in the spirit of the Reviewer's thoughtful comment.

Moreover, any validation of these associations is entirely descriptive in nature, and I am not even 100% sure the authors have taken the care to make sure these associations do in fact make sense. As a case in point, in the BRCA dataset the top component was enriched for genes involved in cell adhesion, proteinaceous extracellular matrix, collagen binding, which qualitatively correlates with the very different changes in tissue architecture between tumor and normal. OK, on the surface this seems promising (although again, completely expected), but do changes in the individual genes and/or classes make biological sense? It is loss of cell-cell adhesion that correlates with disease? Or is it gain of cell-ECM adhesion? Looking at ECM-associated genes becomes much more complicated. Cancerous tissue does not, as the authors seem to imply, arise solely through a loss of normal ECM, but rather a complete reorganization of ECM that is accomplished through changes in ECM proteins, proteases, and ECM-binding proteins. So do the changes in ECM related genes actually reflect this? The authors should also be very careful that a handful of genes is not driving multiple category enrichments.

Answering these questions is important because I think it validates the method, but also may provide some mechanistic insight which is desperately needed here.

We absolutely agree. We have spent time working with Phil Branton, the pathologist involved in the GTEx project, to discuss these images and the significance of the genes found with the images at the extreme ends of the factor values. This collaboration has catalyzed a much more thorough discussion of the significance of these GO terms along with the connection to the biological significance of the pathology images that was lacking in the original manuscript. We also changed the image feature QTL analysis to test for QTLs within a specific cell type, and we believe the interpretation is much more clear in this context.

Although I do not expect the authors to do everything here, it would be nice if they could show that they can identify genes or targets in a way that I would believe this method would have utility in the future both in fundamental research or in the clinic.

I would suggest the authors try digging deeper into the results of their analyses, and in particular the results of the BRCA dataset, as I believe that is where they are most likely to find something insightful. As a suggestion they may consider using methods alternative to GO enrichment to identify genes/proteins and then look at their output on a gene-by-gene basis.

We hope we have done this now, both by developing a very thorough downstream analysis of the image components for sparse latent variable modeling and also by spending much more time in the analysis of these components with pathologists in the revised manuscript.

2) Normal versus graded cancers

Are the authors able to identify only the strongest signals due to the fact that they have included so much normal tissue, especially in the LPP and GTEx analyses? And are often identifying classifying normal versus cancer tissue, instead of low grade vs. high grade?

This is an important point: our goals here were exploratory in nature and not predictive or discriminative. We trained the CAE on healthy samples for healthy tissues, and tumor plus healthy samples for tumor data sets. Because our goals here were not predictive, we only used labels in the training for identifying sets of features; none of the supervised analyses were used for downstream results. Because we had so few image samples, we did not attempt to train a classifier for grade prediction. We added two sentences to the Results section to explain this focus in the context of our methods more explicitly.

3) QTL analysis I do find the attempt to identify associated QTLs VERY interesting, but very preliminary. Unlike some of the results of the correlated gene expression analysis, it is completely if those genes have any meaningful role to play in cancer biology.

We agree, and we believe that this might be the most interesting aspect of the paper. We have redone this analysis in GTEx after making our pipeline more stringent, and we have found some exciting associations in colon images and in thyroid images that we share in the manuscript, both with strong relationships to cancer through the mediating gene. We are eager to get your opinion on these new results. We have also added a Supplemental table with all of the results from the image feature QTLs included (Supplementary Table 9).

4) Other

For a general journal such as Nature Communications, I found this manuscript not easy to go through (and I am in this area). To make it accessible to a broad audience. authors should spend considerable effort changing the figures and text.

Some notable phrases in particular:

"Canonical correlation analysis (CCA) finds linear mappings for two sets of variables observed in paired samples onto a shared latent subspace upon which they are maximally correlated with each other."

"We interpret the latent semantics of the CCA components by examining the enriched molecular functions and tissue specificity of the subsets of genes and also the differences in cellular morphology of the extreme values of the images."

"We then used the activations at the last hidden layer of the aggregate network as the supervised image features in the downstream CCA in place of the image features from the unsupervised CAE."

Thank you for these comments. We agree with you and appreciate the suggestions. We have restructured these sentences and many others to conform to the language and the phrasing common to Nature Communication articles and to avoid unnecessary jargon. We have also moved substantial methodology to the Methods section and out of the Results section.

Reviewer #3

The authors propose and test the combination of two machine learning approaches for inferring gene expression patterns that are associated with histology images. First, using CAE, they capture high-dimensional imaging features, and subsequently they use CCA to jointly infer latent variables from imaging features and gene expression data that co-vary. They apply this approach in varied setting, including data from TCGA and GTEx. They show the existence of expression latent variables that are correlated with histology features.

Overall the method presented combines several interesting approaches with desirable properties, including automatic feature generation from histology images, and accounting for broad patterns that may be related to confounding. However, I have two primary concerns: a) I did not get a good sense of biological usefulness/significance of the method, as in the results were not placed in the context of previous knowledge and finding, b) the statistical significance of the specific correlations reported is not assessed (see below).

We thank the reviewer for this thorough summary. We have added more background and related work to the manuscript to address the first concern. The second concern is addressed below.

- The authors report some imaging based patterns that correlate with expression latent variables, however, there is no discussion of what new biology has been learned (besides the generic GO enrichment analysis) and how it compares with what has been known.

We agree. As noted in the response to Reviewers 1 and 2, we have substantially increased the biological interpretation of these components, both in developing a statistical tool to post-process the components much more thoroughly, and also in working with the GTEx pathologist to understand the pathological images at much finer detail. We now have better statistical significance attached to the findings based on permutations and association testing, and enrichment analysis as well. For complete details, see responses to Reviewers 1 and 2, and also the completely revised manuscript.

- The organization of the articles makes it hard to read: the Results section contains a mixture of materials, methods, results and discussion, but all separated based on dataset. I suggest the authors re-organize the paper's content.

We appreciate the suggestion. We have tried to give sufficient detail about the methods in the Results to make sure that the results can be understood, and we have removed the Methods-type information that can be ignored in order to understand the Results. We still choose to organize the

results in part by data set because jumping from describing the three data sets to then describing results on the three data sets did not allow for concise flow.

- Although there is an overall permutation analysis for assessing the overall significance of features, can the author show that each latent variable is statistically significant? As the authors know, with the high-dimensionality of the data, sCCA will essentially likely find some shared latent dimensions, but which ones are significant?

Latent variable models do not have a null hypothesis (i.e., what would these samples look like without the specific pattern captured in this factor is not well quantified), so there are no clear measures of statistical significance of the components. That said, we have included proportion of variance explained for each component and each study, and we report these in the Results and in a new Supplementary Figure 4. We also include permutation analyses to describe the value of the downstream enrichments in highlighting true biological signal (Supplementary Fig 5).

- How robust is the method with respect to parameter setting for sCCA?

We have added a robustness subsection and an additional Supplemental Figure to address this question. We performed a thorough grid search of the hyperparameters for CCA (Supplementary Fig 2), prioritizing the quality of the reconstruction of the images and gene expression data, and selected the values with optimal reconstruction capabilities. Looking across the complete results in GTEx with different hyperparameters, we find strong correlations among the components (Supplementary Fig 3), indicating robustness across two parameter settings.

- GO enrichment analysis some times include terms with pretty large p-values (e.g., $\sim 10^{-4}$), how many GO terms were analyzed and what's the significance threshold?

We considered all terms in the three categories of the Gene Ontology, and these p-values were uncorrected (as described in the text). However, we agree that some of the p-values are unconvincing, and we have thoroughly revised these in the context of the much deeper downstream analysis of the components. We also select components that have greater enrichment to describe biological significance, and the p-values associated with these components are many orders of magnitude below the correction thresholds.

- The statistical significance threshold for cis-eQTL analysis (FDR 0.1 and FDR 0.2) is pretty large, please justify.

We agree. We have changed the test to be more discriminative and now include a consistent 10% FDR threshold. More generally, though, FDR thresholds represent the proportion of discoveries that are false positives in expectation. Given the number of discoveries we identified, these FDR thresholds are both scientifically reasonable (there are only 10% false positives among our discoveries); this is consistent with our previous GTEx work.

References

- [1] Daniela M Witten, Robert Tibshirani, and Trevor Hastie. A penalized matrix decomposition, with applications to sparse principal components and canonical correlation analysis. *Biostatistics*, page kxp008, 2009.

Reviewers' comments:

Reviewer #2 (Remarks to the Author):

I appreciate the considerable effort the authors have put into revising the manuscript. I still think that this work represents a real methodological achievement. However, my concerns over whether deployment of this method has led to any meaningful impact on our understanding of cancer biology and/or diagnosis remain. I certainly think the associations between gene expression and tissue morphology the authors have found are genuine. Moreover, they have now taken care to describe the processes that may underlie some of these associations. But whether due to the stringency of the method, and/or because of the nature of the datasets they have analyzed (relatively small size? small variance? variability in sample preparation?), I just don't think the authors have found anything useful. It is difficult to assess whether given better datasets, the method could provide more insight; or whether the method is just detecting very strong signals. As a cancer researcher, I don't feel like there is something here that would be worth investigating in more depth. As a diagnostic developer, I don't feel convinced that using this method would now lead to more refined patient stratification.

Whether this gets published in Nature Communications is likely an editorial decision that needs to balance the obvious technical strengths of this work versus the lack of new cancer biology.

As a minor comment, I found the text much more readable than the previous version. But the "Results" section contains far too much description of procedures which are clearly "Methods".

Some examples:

"In particular, the 1024 feature vector was the mean encoded feature value across the 100 image windows of each image; additionally, we whitened this space so that the 1024 features were orthogonal (Supplementary Fig 1)... Using the CAE in an unsupervised approach, the embedding is estimated with the objective of reconstructing the original image as accurately as possible, where the objective is minimizing the ℓ_2 distance between the original and the reconstructed image, using only the estimated 1,024 features..."

"Sparse CCA performs this same projection into a shared latent space, but zeros are encouraged in the projection matrix, identifying small numbers of genes and image features responsible for the variation captured in that component 23..."

"Based on our grid search, we set the sparsity parameter to 0.05 for gene expression observations for BRCA and LGG, and to 0.1 for GTEx; we set the sparsity parameter to 0.15 for the image feature observations across all applications..."

Reviewer #3 (Remarks to the Author):

The authors have addressed my comments. Thank you.

Reviewer #4 (Remarks to the Author):

In my opinion the authors have sufficiently addressed many questions of Reviewer 1. However, a few critical points remain:

% Regarding Reviewer 1, point 6 (taking absolute correlations):

I disagree with the Authors rebuttal and agree with Reviewer 1: By taking absolute correlation values (and thus removing all signs), you are missing out on valuable relationship information. It is true that the signs are relative, and I agree with the authors when they write: "the sign of the loadings can all be swapped". Indeed, you can inverse the signs of the complete dataset, but you can't arbitrarily swap the sign of any SUBSET of correlations (which is what you do when taking absolute values, i.e. you swap the sign only of all the negative correlations). Thus, each sign within the dataset holds value relative to all other signs within that same dataset. And thus you are likely losing valuable relationship information by taking absolute correlations. The authors should SHOW (not just argue) that removing the signs of the correlations does not change the interpretation of their analysis.

% Regarding Reviewer 1, point 8 (biological interpretation, and by extension, validity of the approach):

% the authors at multiple points in the text and rebuttal claim that finding significant gene-set enrichments validates their method. Yet the presentation of those enrichment results is very sparse, with only few - potentially cherry-picked - examples shown/discussed in the main text. I appreciate that the full results are included in the Sup Tables, and browsing them, I find that the complete results look promising. But reading the result section and seeing the figures in its current form I, like other Reviewers for previous versions of the manuscript, was not at all convinced of the biological use of the approach. For the value of this manuscript, therefore, it would in my opinion greatly help if the authors could think of a much more systematic and comprehensive display of the enrichment results for a given dataset, going beyond selected examples and a sup-table. Particularly for the multi-tissue GTEx dataset.

% I am not at all convinced of the linear interpretation of the CAE features that the authors suggest by contrasting one high scoring with one low scoring image. High scoring images should indeed contain a common pattern (e.g. a common tissue), but low scoring images can contain many different patterns. Thus, contrasting a high and low image suggests a false sense of linearity, especially in e.g. a multi-tissue dataset as the GTEx. I see 3 options:

- If the authors want to prove me wrong they could show linearly ranked example images fitting their suggested linear interpretation of the results.

Conveniently, if there is in fact some relevant linear image ranking produced by their CAE features, then showing CAE-feature-sorted example images (similarly to their t-SNE visualization, just on a single image row/sequence) will greatly improve the biomedical interpretation of that particular feature.

- Alternatively, if the authors agree with me and the linear interpretation is flawed, they could acknowledge it does not follow a linear interpretation in writing, and show multiple example low scoring images. Also in this case the suggested linear sorted image visualization would be a better and more comprehensive representation of that CAE feature.

Response to reviewers

Thank you for these thoughtful and thorough reviews for this manuscript. We have addressed each point raised by the reviewers; reviewers' comments are in plain text and *our responses are in blue italic text*. Significant text changes and additions in the main manuscript were marked with **blue text** and large text removals were marked with ~~red-strike-through-text~~. Smaller stylistic or grammar changes were not marked with blue text.

Reviewer #2

I appreciate the considerable effort the authors have put into revising the manuscript. I still think that this work represents a real methodological achievement. However, my concerns over whether deployment of this method has led to any meaningful impact on our understanding of cancer biology and/or diagnosis remain. I certainly think the associations between gene expression and tissue morphology the authors have found are genuine. Moreover, they have now taken care to describe the processes that may underlie some of these associations. But whether due to the stringency of the method, and/or because of the nature of the datasets they have analyzed (relatively small size? small variance? variability in sample preparation?), I just don't think the authors have found anything useful. It is difficult to assess whether given better datasets, the method could provide more insight; or whether the method is just detecting very strong signals. As a cancer researcher, I don't feel like there is something here that would be worth investigating in more depth. As a diagnostic developer, I don't feel convinced that using this method would now lead to more refined patient stratification.

Whether this gets published in Nature Communications is likely an editorial decision that needs to balance the obvious technical strengths of this work versus the lack of new cancer biology.

We thank the Reviewer for thoughtful comments regarding the impact of results presented in this work on cancer biology generally. We hope that by publishing the paper and allowing cancer researchers with much higher quality data than the ones we used to access the methods will lead to the great discoveries that may come from this method applied to specific cancer biological processes. We note that this approach is not meant to stratify patients further, but instead has the potential to uncover classes of tissue- and tumor-morphology related cellular processes that can be used to understand basic tumor biology. We also acknowledge the major problems with the TCGA data sets, where many of the samples show substantial necroptosis, and we are working with researchers at Cancer Research Hospitals to identify better samples for future work.

As a minor comment, I found the text much more readable than the previous version. But the "Results" section contains far too much description of procedures which are clearly "Methods".

Some examples:

“In particular, the 1024 feature vector was the mean encoded feature value across the 100 image windows of each image; additionally, we whitened this space so that the 1024 features were orthogonal (Supplementary Fig 1). . . . Using the CAE in an unsupervised approach, the embedding is estimated with the objective of reconstructing the original image as accurately as possible, where the objective is minimizing the ‘2 distance between the original and the reconstructed image, using only the estimated 1,024 features...”

We moved much of this to the Methods.

“Sparse CCA performs this same projection into a shared latent space, but zeros are encouraged in the projection matrix, identifying small numbers of genes and image features responsible for the variation captured in that component 23...”

While we changed the wording in this sentence, we kept it in the Results as sparsity is essential to understanding the results and interpretations.

“Based on our grid search, we set the sparsity parameter to 0.05 for gene expression observations for BRCA and LGG, and to 0.1 for GTEx; we set the sparsity parameter to 0.15 for the image feature observations across all applications...”

Thank you, we moved this to the Methods.

Reviewer #3

The authors have addressed my comments. Thank you.

Thank you.

Reviewer #4

In my opinion the authors have sufficiently addressed many questions of Reviewer 1. However, a few critical points remain:

Regarding Reviewer 1, point 6 (taking absolute correlations):

I disagree with the Authors rebuttal and agree with Reviewer 1: By taking absolute correlation values (and thus removing all signs), you are missing out on valuable relationship information. It is true that the signs are relative, and I agree with the authors when they write: “the sign of the loadings can all be swapped”. Indeed, you can inverse the signs of the complete dataset, but you can’t arbitrarily swap the sign of any SUBSET of correlations (which is what you do when

taking absolute values, i.e. you swap the sign only of all the negative correlations). Thus, each sign within the dataset holds value relative to all other signs within that same dataset. And thus you are likely losing valuable relationship information by taking absolute correlations. The authors should SHOW (not just argue) that removing the signs of the correlations does not change the interpretation of their analysis.

Yes, you make a compelling point here that I think we did not fully appreciate or address in the concerns from R1 in the first revision, and we address this in the current revision. To address this point, we replaced Figure 4 in the main manuscript with a new version of the heatmap where Pearson's correlation is plotted (not absolute value Pearson's correlation); to ease the study of this new heatmap, we also inverted the signs of some components to ensure that the Pearson's correlation of all components with Chest Incision Time were positive. As the Reviewer notes, this allows the negative and positive correlations within a single component to be studied, but removes no other identifiable information.

We have added the following text to the Results to discuss these findings. "Many of the components capture variation in features correlated with the same subset of covariates, primarily surrounding type of death, sample ischemic time, and gender, age, and weight. Moreover, despite the sets of genes in each component being mostly different, we note that the correlation signs (i.e., positive or negative correlations) are, for the most part, consistent within the covariates across the CCA components. In other words, across these components, e.g., Gender and Ischemic Time have opposite correlation signs. This consistency implies that there are a large number of gene sets that are involved in the same biological processes of postmortem decay with signatures in tissue morphology."

Regarding Reviewer 1, point 8 (biological interpretation, and by extension, validity of the approach):

* the authors at multiple points in the text and rebuttal claim that finding significant gene-set enrichments validates their method. Yet the presentation of those enrichment results is very sparse, with only few - potentially cherry-picked - examples shown/discussed in the main text. I appreciate that the full results are included in the Sup Tables, and browsing them, I find that the complete results look promising. But reading the result section and seeing the figures in its current form I, like other Reviewers for previous versions of the manuscript, was not at all convinced of the biological use of the approach. For the value of this manuscript, therefore, it would in my opinion greatly help if the authors could think of a much more systematic and comprehensive display of the enrichment results for a given dataset, going beyond selected examples and a sup-table. Particularly for the multi-tissue GTEx dataset.

We appreciate this comment and agree that, in aggregate, "the complete results look promising." We have gone through related literature to determine how others have displayed these results in a systematic and comprehensive way, and we thought about it ourselves for a long time, but could not come up with an approach that did not have major flaws akin to the standard cherry picking flaws the Reviewer points out. We note that the last author actually wrote her thesis on statistical models applied to the Gene Ontology, and the amount of thought put into this is substantial and

over a 19 year period. We unfortunately have to defer a response until better ideas are identified.

* I am not at all convinced of the linear interpretation of the CAE features that the authors suggest by contrasting one high scoring with one low scoring image. High scoring images should indeed contain a common pattern (e.g. a common tissue), but low scoring images can contain many different patterns. Thus, contrasting a high and low image suggests a false sense of linearity, especially in e.g. a multi-tissue dataset as the GTEx. I see 3 options: - If the authors want to prove me wrong they could show linearly ranked example images fitting their suggested linear interpretation of the results. Conveniently, if there is in fact some relevant linear image ranking produced by their CAE features, then showing CAE-feature-sorted example images (similarly to their t-SNE visualization, just on a single image row/sequence) will greatly improve the biomedical interpretation of that particular feature. - Alternatively, if the authors agree with me and the linear interpretation is flawed, they could acknowledge it does not follow a linear interpretation in writing, and show multiple example low scoring images. Also in this case the suggested linear sorted image visualization would be a better and more comprehensive representation of that CAE feature.

Thanks to the Reviewer for pointing this out, and we believe that this is the result of a misunderstanding that can be cleared up quickly. The "low scoring images" are actually not images with scores near 0, but instead images with high magnitude negative scores, and as such have identical semantics to the "high scoring images." We do agree, however, that the most extreme samples on the linear ordering are not necessarily indicative of a single tissue type, but they do represent the greatest contrasts in the CAE feature, instead of a high magnitude set versus a low magnitude set as our current wording suggests. We clarified this point in a few places in the revised manuscript. With this additional clarification, we believe we have taken Option 2 suggested by the Reviewer here, which is effectively a relevant linear image ranking produced by the CAE features.

REVIEWERS' COMMENTS:

Reviewer #4 (Remarks to the Author):

I am disappointed to read that the authors have not come up with a better way to present their results, as this leaves unaddressed the main concern regarding the lack of impact of the biological results.